# Autonomous Unmanned Heterogeneous Vehicles for Persistent Monitoring

**Vaios Lappas** [1,*], **Hyo-Sang Shin** [2], **Antonios Tsourdos** [2], **David Lindgren** [3], **Sylvain Bertrand** [4], **Julien Marzat** [4], **Hélène Piet-Lahanier** [4], **Yiannis Daramouskas** [5] **and Vasilis Kostopoulos** [5]

[1] Department of Aerospace Science & Technology, National & Kapodistrian University of Athens, 34400 Athens, Greece

[2] Centre for Cyberphysical and Autonomous Systems, School of Aerospace, Transport and Management, Cranfield University, Cranfield MK43 0AL, UK; h.shin@cranfield.ac.uk (H.-S.S.); a.tsourdos@cranfield.ac.uk (A.T.)

[3] Swedish Defence Research Agency FOI, 90621 Umea, Sweden; david.lindgren@foi.se

[4] Traitement de l'Information et Systèmes, ONERA, Université Paris Saclay, 91190 Palaiseau, France; sylvain.bertrand@onera.fr (S.B.); julien.marzat@onera.fr (J.M.); helene.piet-lahanier@onera.fr (H.P.-L.)

[5] Applied Mechanics Lab, Department of Mechanical Engineering and Aeronautics, University of Patras, 26504 Patras, Greece; daramousk@ceid.upatras.gr (Y.D.); kostopoulos@upatras.gr (V.K.)

\* Correspondence: valappas@aerospace.uoa.gr

**Abstract:** Swarms of unmanned vehicles (air and ground) can increase the efficiency and effectiveness of military and law enforcement operations by enhancing situational awareness and allowing the persistent monitoring of multiple hostile targets. The key focus in the development of the enabling technologies for swarm systems is the minimisation of uncertainties in situational awareness information for surveillance operations supported by 'system of systems' composed of static and mobile heterogeneous sensors. The identified critical enabling techniques and technologies for adaptive, informative and reconfigurable operations of unmanned swarm systems are robust static sensor network design, mobile sensor tasking (including re-allocation), sensor fusion and information fusion, including behaviour monitoring. The work presented in this paper describes one of the first attempts to integrate all swarm-related technologies into a prototype, demonstrating the benefits of swarms of heterogeneous vehicles for defence applications used for the persistent monitoring of high-value assets, such as military installations and camps. The key enabling swarm system technologies are analysed here, and novel algorithms are presented that can be implemented in available COTS-based unmanned vehicles. The algorithms have been designed and optimised to require small computational power, be flexible, be reconfigurable and be implemented in a large range of commercially available unmanned vehicles (air and ground).

**Keywords:** unmanned systems; persistent monitoring; autonomy

## 1. Introduction

Advances in microelectronics, UAV development, autonomous systems and guidance, navigation and control systems have enabled the development of unmanned vehicles to perform complex missions, such as surveillance and persistent monitoring tasks. These autonomous systems, mostly operated in small groups, are able to deliver significant amounts of data in real time; however, they are constrained by the levels of autonomy available and the difficulties of integrating multiple vehicles in swarms. Multiple unmanned vehicles can add not only strength in numbers, but unique capabilities in redundancy, mission flexibility and target tracking/monitoring which can enhance mission capabilities for defence and law enforcement needs. Asymmetrical warfare and the need to simultaneously detect unidentified targets with multiple behaviours are challenging current autonomous systems, as a single, yet capable UAV is not necessarily able to detect, track and persistently monitor

multiple targets. Furthermore, large conventional unmanned vehicles, or UAVs, do not have advanced levels of autonomy to date, but rather use human-in-the-loop decision making and control protocols and resources to perform surveillance and monitoring tasks which might become fully automated. Our paper presents the results of a recently completed research project in which a pilot scaled autonomous swarm of unmanned vehicles performed a persistent monitoring mission to protect a mock high-value asset, such as a military installation. A fully integrated, autonomous swarm-based framework was developed and simulated, in which sensor fusion, behaviour monitoring, target and resource allocation and guidance/control algorithms facilitate efficiently tracking hostile targets. A scaled outdoor demonstration using 7–10 UAVs and UGVs has shown the feasibility, challenges and benefits of using a swarm of unmanned vehicles for defence missions. The paper presents an overview of the mission scenario addressed and the swarm system architecture developed. Section 3 details the swarm tracking and control strategy developed. Section 4 details the situational awareness and sensor fusion algorithms developed for implementation in unmanned aerial and ground vehicles which have limited computational power. Section 5 describes the guidance laws developed for the swarm, and Section 6 presents the fully integrated swarm system that was successfully implemented on multiple unmanned vehicles in an outdoor environment (Cranfield Airport) and used to provide persistent monitoring of a high-value asset.

## 2. Swarm System Architecture

The work presented in this paper was focused on testing and demonstrating that efficient and effective operation of unmanned swarm systems can bring a profound impact to the military arena. The key focus in the development of the enabling technologies was the minimization of uncertainties in situational awareness information for surveillance operations supported by a swarm system of systems composed by static and mobile heterogeneous sensors. The functionalities and features of the main enabling technologies developed can be summarized as follows: (i) an optimal sensor network—static sensors to sense the environment and potential targets; (ii) mobile tasking, including decision making, assigning mobile sensing platforms to the set of tasks, completing the situational awareness information gap on the sensor network and improving the searching and monitoring capabilities; (iii) sensor fusion, target detection and identification, improving the accuracy of the target tracking performance; (iv) information fusion, behaviour monitoring and target assessment (obtaining the threat levels of the targets), to improve the decision making of the mobile sensing platforms; (v) cooperative guidance, path planning and following mobile sensing platforms. Research on all building blocks of a swarm system is extensive [1–15] (task allocation, trajectory planning, etc.), and various new techniques can be applied to swarm scenarios, such as for nonlinear trajectory estimation [15].

*Swarm Mission Scenario, Functionalities and Requirements*

A mission scenario is used in the work in which a specific area of high interest requires persistent monitoring/surveillance. It is assumed that the scenario takes place near a battlefield in a conflict with a well-armed and competent opponent. The high-value asset (HVA), a military camp, is located in a rough and partly hilly terrain, and a sophisticated sensor system to support perimeter surveillance is available. As the terrain limits the visibility in the protected area, centralized sensors are ineffective. Instead, ground sensors are distributed in a large area around the camp that facilitate early indications of enemy reconnaissance or approaching formations. The ground sensors are sensitive to the presence of humans, vehicles and animals and give prompt alarms if potential targets are in the vicinity. The ground sensors do little to assess the nature and severity of the threats but are on the other hand robust and persistent. They enable coverage of a large area for their price and maintenance requirements. To support alarm verification, the network of ground sensors is completed with drones (air and ground) that operate autonomously in the sensor system. The drones are only activated for special tasks and missions. For instance, on a

ground sensor alarm, the drones do a verification that may give the guard high fidelity video and other sensor data from the target or target area in real time. The drones are autonomous and can collaborate in pairs, groups or swarms to meet the high demands on service quality and persistence toward hostile means of deception and attacks.

The key top level functional requirements in the HVA mission scenario are presented in Table 1.

**Table 1.** Key top level functional requirements in the HVA mission scenario.

| Num. | Functional Requirement | Vehicles | Time (min) | Performance Metrics |
|---|---|---|---|---|
| 1 | Swarm performs persistent monitoring in an indoor environment 20 × 5 × 3 m with 2 targets | 1 UGV 2 UAV | ~20′ | Position accuracy, speed, situational awareness, path planning accuracy, target position accuracy, speed, time to track, loss of data, link robustness, resilience, bandwidth |
| 2 | Swarm performs persistent monitoring of a 500 × 500 × 100 m area with multiple targets | 2 UGV 5 UAV | ~20–30′ | |

### 3. Swarm Sensor Tasking and Control

The task allocation problem is assigning each agent (mobile platform and/or onboard sensor) to a task or a bundle of tasks. The strength of autonomous swarm systems of aerospace vehicles hinges on the distributed nature of the resources available, making the successful assignment of these resources key to maximize its operational advantages and thus minimizing uncertainties.

Efficient cooperation of a swarm of autonomous systems, termed as task allocation, is a vital part for mission success. Task allocation problems are defined as reward function maximization problems. The main objective of task allocation algorithm is to find out the agent and task combinations which maximize the reward function. For the problems with small numbers of agents and tasks, it is possible to calculate the reward function values of all the possible agents and tasks combinations and select the combination with the maximum reward function value. However, since task allocation problem is combinatorial and NP hard, as the numbers of agents and tasks enlarge, computation loads dramatically increase. In this report, approximate algorithms are applied for task allocation problems. One advantage of approximate algorithms is that the computation load is mathematically calculated. This implies that the required computation load for the given task allocation problem and optimization algorithm. The other one is that, although approximate algorithms cannot guarantee the actual optimal solution, they provide solutions with mathematically guaranteed certain levels of optimality.

It is evident that the key enabler of the task allocation is allocation with near real time, so that the agents in the swarm system should be immediately allocated to appropriate task(s). To this end, this project will develop an approximation. Remind that approximation algorithms balance between the optimality and the computational time. Moreover, their quality of the solution and polynomial time convergence could be theoretically guaranteed as long as the objective function satisfies certain conditions, e.g., submodularity. The first focus will be to design a new task allocation model in a manner guaranteeing the submodularity. Note that the objective function of the task allocation problem in the project will be the situational awareness information. It is well-known that the information generally holds the submodularity. Hence, it will be possible to design the problem to hold the submodularity condition. Once the submodularity of the new task allocation model is proven, then implementation of such an approximate algorithm will enable resolution of the task allocation problem in an almost optimal manner in real time. A novel task allocation algorithm based on the greedily excluding technique was developed and validated for EuroSWARM which was shown to be more computationally efficient than current algorithms and can enable the use of existing, low power COTS processing technology available in the UAV and microelectronics markets.

Recently, there have been scientifically interesting developments in approximation algorithms for submodular maximization subject to abstract matroid constraints. In this section a novel decentralized task allocation algorithm has been developed for swarms which consist of for Multi Robot Systems (MRS) using approximation guarantees for general positive-valued submodular utility functions. Two approximate algorithms are investigated and enhanced for swarm control. The well-known greedy algorithm was analysed and enhanced as a greedily excluding algorithm. In the newly developed algorithm, in contrast with other submodular maximization algorithms, at the beginning, all the tasks are assigned to each agent. In each step, the new algorithm reward function reduces, by 'excluding' all subtasks of each task (which are calculated), and then the task with the smallest reward function reduction is then excluded. These procedures are repeated until each of all of the tasks is assigned to a single agent. The main purpose of introducing greedily excluding algorithm is to relieve the computation load of the task allocation algorithm. The two major criteria of performance validation on task allocation algorithms are the level of guaranteed optimality and computation load, as mentioned above. The task allocation algorithms are required to be operated in real time for rapidly changing environments, such as those encountered in battlefield scenarios. However, in cases with large numbers of tasks, the computation loads dramatically increase. This implies that low computation load is a major requirement for the application of task allocation algorithms to rapidly changing problems with many tasks. In the proposed task allocation algorithm, the computation load is reduced using a greedily excluding algorithm. This computation load reduction capability is mathematically calculated, and it is shown that the reduction grows as the number of tasks enlarges. The optimality of the greedily excluding algorithm is tested and compared with the greedy and exhaustive algorithms through simulation.

The Greedy algorithm (Figures 1 and 2) is one of the most well-known submodular maximization algorithms. The element which provides the largest marginal gain is selected and added to the solution set. The selected element is excluded from the ground set. The same procedures are repeatedly conducted while predefined constraints are satisfied. The greedy algorithm under cardinality constraint could be expressed as below (Algorithm 1).

---

**Algorithm 1** Greedy Algorithm with Cardinality Constraint

---

1: $A \leftarrow \varphi$
2: while $|A| < k$ do
3: $\quad$ $e^* \leftarrow \text{argmax}_{e \in G} f_A(e)$
4: $\quad$ $A \leftarrow A \cup \{e^*\}$
5: $\quad$ $G \leftarrow G \backslash \{e^*\}$
6: end
7: return $A$

---

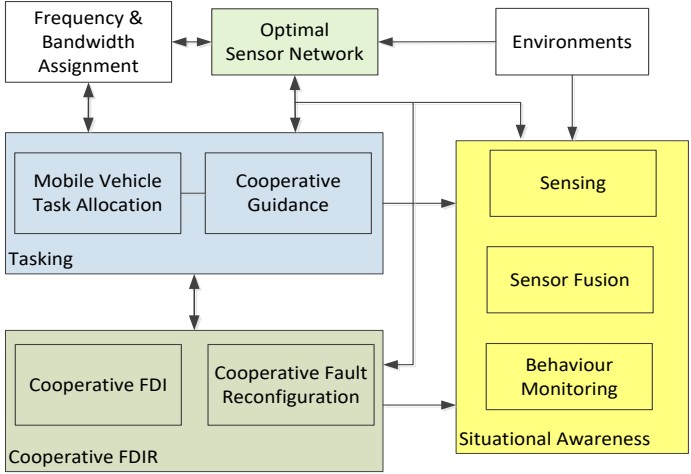

**Figure 1.** An overview of the autonomous swarm framework building blocks.

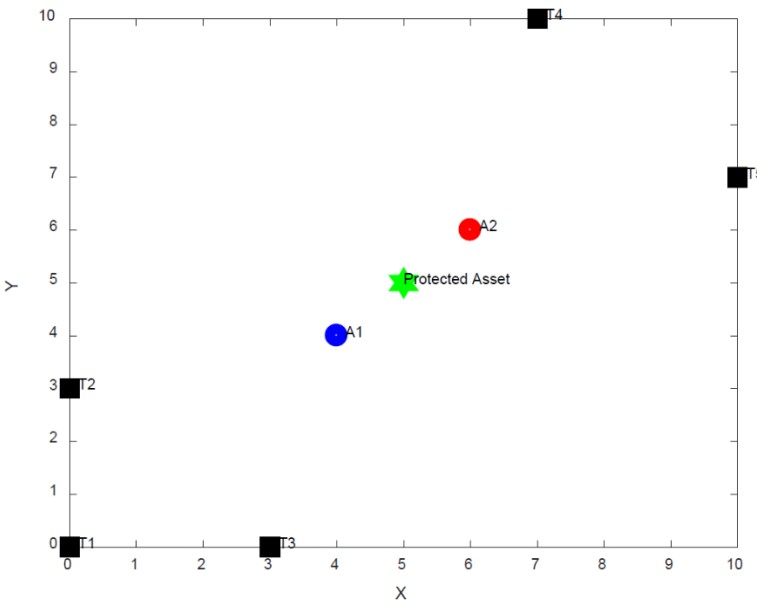

**Figure 2.** Swarm task allocation with multiple UAV agents and targets.

*A* is the solution set and *G* is the ground set, *k* is the cardinality limit of the solution set and $f(\cdot)$ is the reward function. As shown in the above figure, the algorithm terminates when the number of elements of the solution set reaches the cardinality limit. The task allocation algorithm proposed in this section is based on the basic greedy algorithm. In each step, the marginal reward function values for all the possible agent and task combinations. The agent and task combination with the largest marginal gain is selected and this task is assigned to the solution set of that agent. The assigned task is excluded from the ground set of tasks. This algorithm is described in Algorithm 2.

---

**Algorithm 2** Task Allocation Algorithm using Greedy Algorithm

---

1: $A_1, \ldots, A_N \leftarrow \varphi$
2: while $|T| > 0$ do
3:     $e^*, i^* \leftarrow \text{argmax}_{e \in T, i \in [1, N]} f_{Ai}(e)$
4:     $A_{i*} \leftarrow A_{i*} \cup \{e^*\}$
5:     $T \leftarrow T \backslash \{e^*\}$
6: end
7: return $A_1, \ldots, A_N$

---

Where $Ai$ is the solution set of agent *i* and N is the number of agents, *T* is the ground set of tasks. As described in the above figure, this algorithm assigns one task at each step. After all the tasks are assigned, the algorithm terminates.

The Greedily excluding algorithm is introduced and used as the swarm task allocation algorithm and builds on the previously presented algorithms shown in Figures 1 and 2. In this algorithm, all the elements of the ground set are assigned to the solution set. In each step, the element whose exclusion results in the smallest reward function value reduction, is excluded, from the solution set. This procedure is repeated until a certain constraint is satisfied. Here, the cardinality constraint is implemented as shown in Algorithm 3.

---

**Algorithm 3** Greedily Excluding Algorithm with Cardinality Constraint

---

1: $A \leftarrow G$
2: while $|A| > k$ do
3:    $e^* \leftarrow \text{argmin}_{e \in A}, [f(A) - f(A - \{e\})]$
4:    $A \leftarrow A \backslash \{e^*\}$
5: end
6: return $A$

---

The proposed task allocation approach using the Greedily excluding algorithm is shown in Algorithm 4.

---

**Algorithm 4** Task Allocation Algorithm using Greedy Excluding Algorithm

---

1: $A_1, \dots, A_N \leftarrow T$
2: $A_1^*, \dots, A_1^* \leftarrow T$
3: while each of the tasks is assigned to a single event, do
4:    $e_i^* \leftarrow \text{argmin}_{e \in A^*}, [f(A) - f(A - \{e\})]$ for $i \in [1, N]$
5:    $A_i \leftarrow A \backslash \{e_i^*\}$
6:    $A_i^* \leftarrow A_i^* \backslash \{e_i^*\}$
7:    for all the tasks
8:      if a task $e'$ is not assigned to any agent,
9:        $k^* \leftarrow \text{argmax}_k[f(A_k + \{e'\}) - f(A_k)]$ where $k \in \{\}i \mid e_i^* = e'\}$
10:       $A_{k^*} \leftarrow A_{k^*} + \{e^*\}$
11:    end
12: end
13: end
14: return $A_1, \dots, A_N$

---

The reward function consists of two parts. The first part is the function which shows whether the targets are matched with the proper sensors. In this reward function, the visibility of the *j*-th target, $\lambda_j$, is assumed to be defined as a simple Gaussian distribution with mean and variance.

$$\lambda_j = N\left(\mu_j, \sigma_j^2\right) \tag{1}$$

The sensing characteristics of *i*-th sensor, are also defined as a simple Gaussian distribution:

$$\nu_i \sim N\left(\mu_i, \sigma_i^2\right) \tag{2}$$

Bhattacharyya distance between the probability distribution functions of *i*-th agent and *j*-th target is defined as:

$$\Delta(\lambda_i, \nu_i) = \frac{1}{8}(\mu_j - \mu_i)^T \left(\frac{\sigma_j^2 + \sigma_i^2}{2}\right)(\mu_j - \mu_i) + \frac{1}{2}ln\left(\frac{(\sigma_j^2 + \sigma_i^2)/2}{\sqrt{\sigma_j^2 \sigma_i^2}}\right) \tag{3}$$

Using Bhattacharyya distance, the relevancy between *i*-th agent and *j*-th target $\omega_{ij}$ is obtained:

$$\omega_{ij} = e^{-\frac{\Delta^2(\lambda_j, \nu_i)}{2\sigma^2}} \tag{4}$$

where $\sigma$ is a design parameter. In order to limit the relevancy between agent and task set, the reward function on sensor suitability, $f^1$, is developed as shown below:

$$f^1(A_i) = min\{M_i(A_i), \delta M_i(T)\} \tag{5}$$

where $\delta \in [0, 1]$ is a design parameter and $T$ is the ground set of tasks. The second part of the reward function is to prevent assigning excessively many tasks to a single or small part of agents. This reward function, $f^2$, for $i$-th agent is defined as follows:

$$f^2(A_i) = \sum_{j=1}^{|A_i|} \mathrm{I} - \frac{j}{|T|} \tag{6}$$

where $A_i$ is the set of tasks assigned to $i$-th agent and $T$ is the ground set of tasks. Note that the sum of $f^2$ for all agents diminishes as the tasks are concentrated on a single agent. The total reward function, $f^{TOT}$, for $i$-th agent is defined as the weighted sum of those two partial reward functions, $f^1$ and $f^2$:

$$f^{TOT}(A_i) = a_1 f^1(A_i) + a_2 f^2(A_i) \tag{7}$$

where $\alpha_1$ and $\alpha_2$ are the weights on each part, and they are defined by the users according to the problem cases. The total reward function for the task allocation result is defined as the sum of $f^{TOT}$ for all the agents.

$$f(A_1, \ldots, A_N) = \sum_{i=1}^{N} f^{TOT}(A_i) \tag{8}$$

*Swarm Task Allocation Numerical Simulation*

In order to demonstrate the performance of the proposed task allocation algorithm, a simulation of a swarm system of aerial and ground unmanned vehicles is constructed, which consists of 3 enemy soldiers and 2 enemy vehicles which approach a protected high-value asset (camp), and the positions and visibility of the targets defined in Table 2 and Figure 1. 2 UAVs detect the targets in this case. One UAV is equipped with an infrared sensor, and the other UAV has an optical sensor.

**Table 2.** Swarm task allocation simulation parameters.

|  | $j$ | $(X_{Ti}, Y_{Ti})$ | $\mu_j$ | $\sigma_j^2$ |
|---|---|---|---|---|
| Soldiers | 1 | (0, 0) | 1 | $1^2$ |
|  | 2 | (0, 3) | 1 | $1^2$ |
|  | 3 | (3, 0) | 1 | $1^2$ |
| Vehicles | 4 | (7, 10) | 10 | $5^2$ |
|  | 5 | (10, 7) | 10 | $5^2$ |
|  | $i$ | $(X_{Ai}, Y_{Ai})$ | $\mu_i$ | $\sigma_j^2$ |
| Infrared Camera | 1 | (4, 4) | 1 | $1^2$ |
| Optical Camera | 2 | (6, 6) | 10 | $5^2$ |

The design parameters for the reward function selected are $(\sigma, \alpha1, \alpha2) = (1, 0.5, 0.5)$. In the simulation mission scenario, moving targets are approaching a protected asset, and the agents with sensors are operated to track and observe the targets. The initial positions of the targets are assumed to be known from external sensors, and the task allocation is autonomously. The main objective of the task allocation, as shown in Figure 2, is to assign targets to the agents equipped with IR/optical sensor which fits the characteristics of them most. Additionally, the tasks should not be assigned excessively on a certain agent; they should be assigned as evenly as possible to the agents. Two sorts of moving targets are considered in this simulation study. The first group is the enemy soldiers, which emit a heat signature. The second group is the enemy ground vehicles, whose purpose differs from their equipment. Two types of UAVs are considered to detect mobile targets. Some of the UAVs are equipped with an infrared sensor (IR), which is suitable for sensing heat emitted from human bodies. The other UAVs are equipped with optical sensors.

The task allocation results for the swarm simulation are obtained using the task allocation formulation of Equations (1)–(7).

The task allocation in results, Table 3, show that the computation loads of the newly developed task allocation approximate algorithms are significantly less than exhaustive search case while achieving the same level of optimality. The computation load of the newly developed greedily excluding algorithm is smaller than that of greedy algorithm by 40% (number of $f^{TOT}$ calculations). Despite the differences in computation effort, all three of the task allocation algorithms compared show the same task allocation performance output, having assigned all the soldiers to the infrared camera-equipped UAV and all the vehicles to the optical camera.

**Table 3.** Swarm task allocation simulation results.

| Algorithm | Task Allocation Result | # of $f^{TOT}$ Calculation | $f(A_1,\ldots,A_N)$ | Optimality (%) |
|---|---|---|---|---|
| Greedy | $A_1 = \{1,2,3\}$ $A_2 = \{4,5\}$ | 60 | 4.1 | 100 |
| Greedy Excluding | $A_1 = \{1,2,3\}$ $A_2 = \{4,5\}$ | 36 | 4.1 | 100 |
| Exhaustive (Optimal) | $A_1 = \{1,2,3\}$ $A_2 = \{4,5\}$ | $3^5$ (=243) | 4.1 | 100 |

The task allocation algorithms using approximate submodular maximization and compared for swarm type of scenarios have been selected due to their ability to handle multiple agents. In order to decrease the computation load of the well-known greedy algorithm, a greedily excluding algorithm was developed. Through simulation and using different mission scenarios (number of agents/targets), it is shown that the computation load of the newly developed greedily excluding algorithm is smaller than that of greedy algorithm, and the difference increases as the number of tasks becomes larger. The task allocation problem for target detection was set up and simulated using the greedy and greedily excluding algorithms. The results are compared with the actual optimal solution, which requires much larger computation load. The simulation results show that the computation loads are smaller in the greedily excluding algorithm, but the task allocation results are the same with the optimal solution. The proposed task allocation algorithm will thus enable the use of existing, low power COTS processing technology available in the UAV and microelectronics markets for use on swarm scenarios for defence applications which use multiple unmanned vehicles and targets.

## 4. Situational Awareness

Compared to traditional traffic surveillance sensors such as loop detectors and video cameras positioned at fixed locations, UAV aerial sensing can provide better coverage with the capability to survey large areas at a high speed without being confined to prescribed ground navigation routes. Therefore, this airborne monitoring capability enables suspicious or unusual behaviour in road traffic or on the battlefield to be identified and investigated promptly so that operator's situational awareness is increased in support of border patrol, law enforcement and protecting infrastructure. Typical behaviour monitoring or abnormal detection approach requires domain experts to analyse the obtained data to detect the potentially suspicious behaviours. This approach is cumbersome and unsustainable under a deluge of data and information which could result from complex scenes. Therefore, there is a strong need to develop high-level analysis algorithms to process target information and detect anomalous behaviours, to reduce the human operator's workload. Behaviour monitoring, or more generally detecting anomalous behaviours, usually can be classified into two routes: the first approach codifies the behaviours using experience and domain knowledge of experts and the behaviours are learned from data in the second approach. A general framework for autonomous behaviour has been developed for the monitoring

of ground vehicles/targets using airborne surveillance to notify the human commander about the potentially dangerous vehicles.

### 4.1. Mathematical Model of Target Monitoring

The proposed approach consists of a trajectory analysis tool and an abnormal driving mode classifier based on Refs. [16,17]. The trajectory analysis tool extracts the driving (target) modes, defined by specific alpha-numeric strings, from the filtered target trajectories using speed and curvature analysis. The driving mode classifier provides potential threat alerts by means of a learning-based string-matching approach. With reference to the pattern matching process, a neural-network based regular expression dictionary (RD) algorithm is introduced to match commonly observed target behaviours. The advantage of utilizing the RD, instead of simple string matching, lies in its flexibility and generality in handling the minor differences between two patterns that perform almost the same behaviours of interest. Simulation results are performed to demonstrate the effectiveness of the proposed framework.

The proposed monitoring method utilises a regular expression dictionary (RD) to match commonly observed target behaviours. The RD contains several bins with each bin representing a set of regular expressions of behaviours that have same regularity level. The advantage of utilising the regular expression, instead of simple string matching, lies in its flexibility and generality in handling the minor differences between two patterns that perform almost the same behaviours of interest. Compared to typical learning-based approaches, especially neural network, the proposed RD approach can be easily adapted to different scenarios without significant changes. This can be carried out by simply moving one string from one bin to another bin or simply adding one specific string to one bin, depending on the application scenarios. Another promising advantage of the proposed RD approach is that it can significantly reduce the computational time by a top-to-down search, compared with purely neural network assessment. The behind reason of this aspect is that most targets are normal and therefore, their corresponding behaviours can be readily matched by the top bins.

The simplest and most intuitive way to implement the pattern matching for abnormal detection is to define a reference RD pattern and match the extracted driving modes with the reference patterns to find particular threat. The advantage of this approach is that it can precisely identify particular behaviours of interest. However, it is clear that this approach requires domain experts to define the reference patterns for specific scenarios and therefore, is case-by-case solution, which is not in a cost-effective manner. To tackle this issue and provide the possibility to detect general unexpected target behaviours that significantly differ from the regular manoeuvres exhibited by the vehicles. In this section, a new learning-based pattern matching approach is used for behaviour monitoring by a swarm of agents/unmanned vehicles.

The proposed approach defines a driving mode $m_k$ at each time instant k. The driving mode characterises the moving behaviour of the monitored target vehicle during the considered time-window. These modes can then be leveraged for defining classes of complex behaviours that could draw the attention on the monitored target. This is achieved by comparing the driving modes, extracted from the filtered vehicle trajectories, with particular pre-defined behaviour strings by the means of pattern/string matching. Considering, for example, the case where the monitoring UAV is interested in detecting a 'deceleration + stop/slowly moving' manoeuvre performed by a vehicle near a protected military base. An example of pattern to look for could be '444,000', meaning that the vehicle decelerates for achieving a velocity that is close to zero or keeping stationary to monitor the military base. Obviously, lots of similar strings could be recognised as this peculiar behaviour, e.g., '944,000', and '994,400'. Therefore, using simple exact string matching might be an excessively strict policy to pattern recognition and cannot account for minor differences in the compared strings. This means that the exact string matching might consider a potentially threat as a normal vehicle. Although one can use thresholding to account for the minor

difference between the referenced patterns and extracted driving mode strings, the tuning of the threshold is often a complex process, requiring deep insight into the considered scenario. These observations motivate the investigation of a new and flexible approach to define the reference patterns. In order to overcome the shortcomings of simple string matching, this report suggests a more flexible way to define the reference patterns by the means of RD. This allows defining a fixed binding structure for the behavioural patterns of interest, specifying mandatory and optional terms.

The objects of the proposed learning process are patterns of driving modes, which are sequences of alphanumerical characters with no ordinal meaning. Given this characterisation, the most suitable learning approach appears to be the use of a neural network. This tool, if properly trained, can accept a sequence of driving modes of fixed length as input and produce a single value denoting to what extent the pattern can be assumed as normal or abnormal. After properly training the neural network, it can be utilised to assess the vehicle trajectories. However, the issues here are the assessment tends to suffer from a high computational burden for real-time applications when considering complex scenarios and the neural network needs to be re-trained when adding some 'assumed normal/abnormal' behaviours to the neural network for some specific scenarios. For example, the 'deceleration + stop/slowly moving' pattern is a normal behaviour when we consider public traffic monitoring but is becomes abnormal when a vehicle is loitering a high-value asset such as a military base. In order to accommodate these issues, a RD dictionary algorithm is created by summarising the observed manoeuvre patterns provided by the neural network. This can be carried out by providing a new dataset, called RD-training set, as an input dataset to the neural network and follows the same procedure used for the NN-training set. Since the output of the neural network lies in [0, 1], the RD-training driving patterns can be divided into several bins based on their associated neural network output as:

$$
\begin{cases}
net(p_i) > \gamma \\
net(p_i) \in \left[ 1 - \frac{(1-\gamma)(j-1)}{N_b}, 1 - \frac{(1-\gamma)j}{N_b} \right]
\end{cases} => p_i \in b_j \tag{9}
$$

where $net\ (\cdot)$ denotes the neural network operator; $p_i$ is $i$-th pattern of the RD-training set; $\gamma$ is the threshold on the neural network output for the pattern to be considered; $N_b$ represents the number of bins of the RD dictionary; and $b_j$ stands for the $j$-th bin. After splitting the RD-training patterns into different bins, a set of regular expressions is then generated for each bin to represent the level of regularity. As we stated earlier, by utilising the regular expressions, one can accommodate the minor difference between two different but almost same patterns. Given a RD dictionary, the assessment can then be performed by searching the dictionary from the top to find the regular expression that matches with the input pattern. Note that most of the driving patterns in a real-world scenario are normal. Therefore, searching from the top bin can save the computational power. Assume that the querying result of a generic input pattern is $q$, then, the normal level of this pattern is given by:

$$
1 - \frac{(1-\gamma)(q-1)}{N_b} \tag{10}
$$

As stated earlier, the advantage of the RD dictionary, compared with the neural network, is that it can be easily updated in a real time. This can be carried out by adding a new pattern into one bin or moving one pattern from one bin to another bin. For example, '44,000' is a normal behaviour pattern when applying to public traffic/driving behaviour monitoring of targets, but it might be a suspicious pattern when considering a high-value asset/military base monitoring scenario. Therefore, when the scenario changes to a military base monitoring, one needs to move the regular expression that represents '44,000' pattern to a bin with the output close to 0. By doing so, the proposed approach can quickly adapt to different scenarios without changing the assessment architecture and thus reducing complexity and computational effort.

### 4.2. Numerical Simulation of Target Monitoring Algorithm

This section validates the proposed approach through numerical simulations. In order to train the neural network, the dataset with positive labels are manually created by the means Markov Chain. By means of the Markov Chain, plausible patterns are generated as the positive data to train the neural network. As for the negative dataset, the pattern is generated as a random sequence of driving modes where a sub-sequence is added that represent a known abnormal behaviour. The following datasets have been produced when applying the proposed algorithm: (i) NN-training set: 20,000 patterns, including 75% positive and 25% negative data, used to train the neural network; (ii) RD-training set: 20,000 patterns, including 75% positive and 25% negative data, used to query the trained network and create the RD dictionary. A simplified military scenario using the high-value asset (military base) baseline shown in Figure 3 is used, to showcase the performance benefits of the proposed behaviour recognition algorithm as shown in Figures 4 and 5. In the considered scenario, there are six different roads, as shown in Figure 4, and the routes or roads of interests are 2, 3, 4, 5 as they are around the military base.

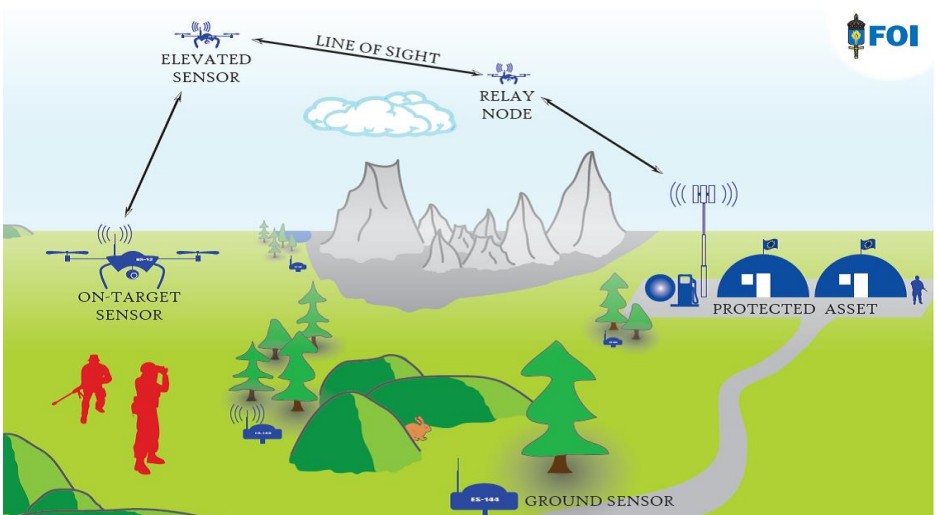

**Figure 3.** Surveillance of a high-value military asset, where a swarm of air/ground drones serve multiple purposes, such as acquisition of completing camera angles and establishing relay chains for effective and secure communication (source FOI).

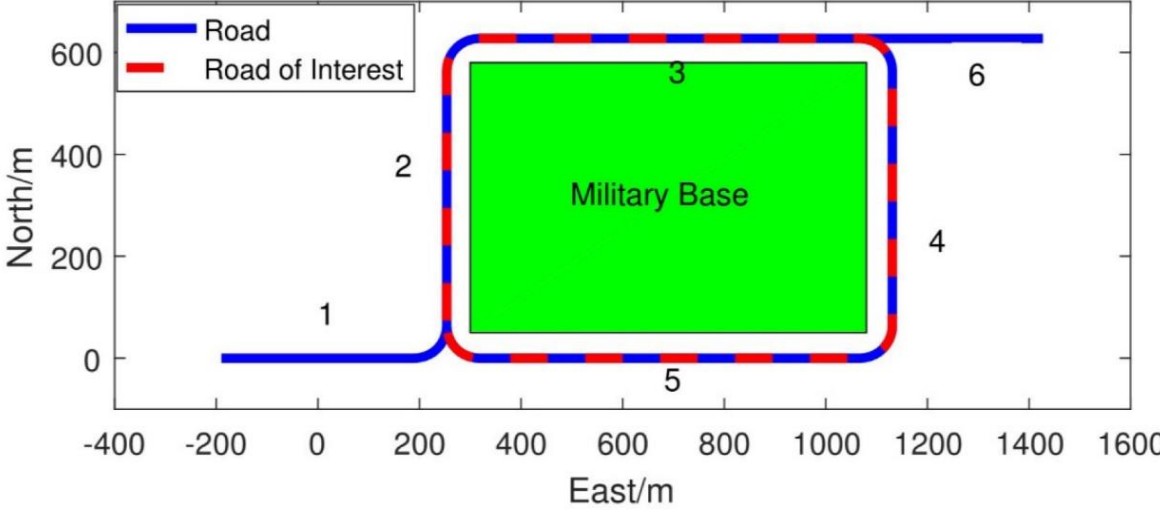

**Figure 4.** Simulation scenario for monitoring a high value asset (military base).

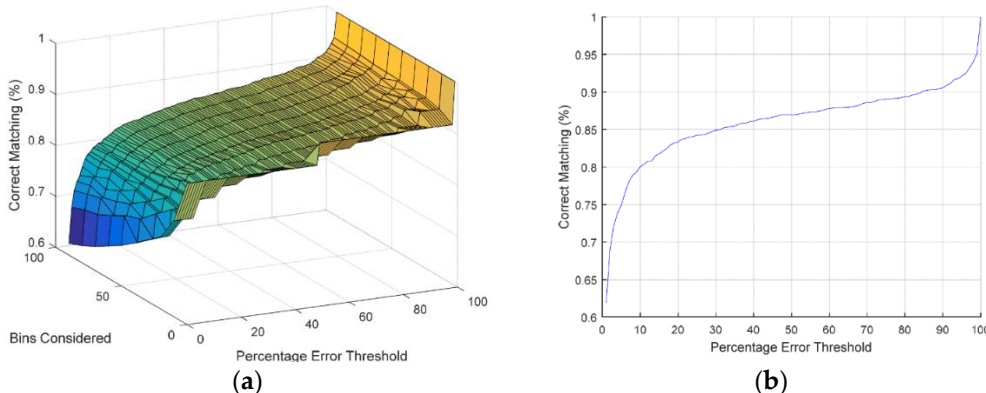

**Figure 5.** (**a**) Results for the RD dictionary using the neural network. (**b**) Results for the RD dictionary using the neural network with 100 bins.

With regard to the accuracy of the generated RD dictionary, as it is generated by the trained neural network, it is reasonable to see to what extent the RD dictionary query output matches with that of the neural network for various test patterns. To assess the accuracy of the algorithm, a test set with 2000 randomly generate patterns is used, for this comparison. The simulation results are shown in Figure 6a,b. The correct matching in the results means that the difference of the outcomes of these two methods lies within a given error threshold. The numeric results indicate that the proposed dictionary query approach has been able to reproduce the NN outputs for a fair portion of the test patterns accounting for approximately 85% of correct matching assuming an error threshold of 20%.

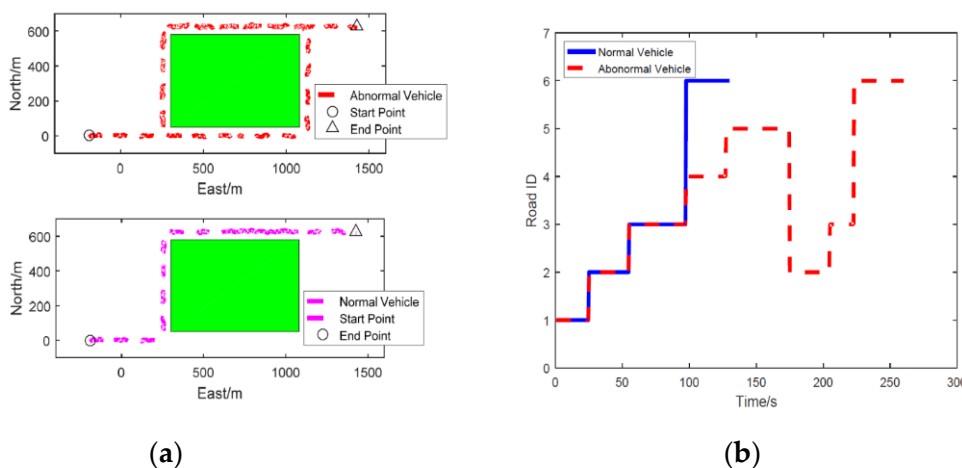

**Figure 6.** (**a**) Assessment for the normal vehicle. (**b**) Assessment for the abnormal vehicle.

Two vehicles, normal and abnormal, with different trajectories are considered. For the abnormal vehicle, it starts from Road 1, loitering the military base through Roads 2, 3, 4, 5, and performs a 'deceleration + stop/slowly moving' manoeuvre at time 145 s at Road 5. For the normal vehicle, it starts from Road 1, regularly pass the military base through Roads 2, 3, and ends at Road 6. The trajectories of these two vehicles are presented in Figure 6a and the road histories are shown in Figure 6b. The assessment results, using neural network and RD dictionary query, of these two vehicles are provided in Figure 7a,b, where Figure 7a is for the normal vehicle and Figure 7b is for the abnormal one. It is clear that the regularity level, provided by both neural network and RD dictionary, of the normal vehicle is very close to 1. This means that the proposed approach considers this car behaves regularly. From Figure 7b, it can be observed that the proposed assessment method successfully identifies the potential threat at around 145 s which is indicated by the output regularity level is close to 0. The recorded running time shows that the RD dictionary query saves

approximate 50% time than that of the neural network method. This clearly verifies the proposed new RD method can reduce the computational time.

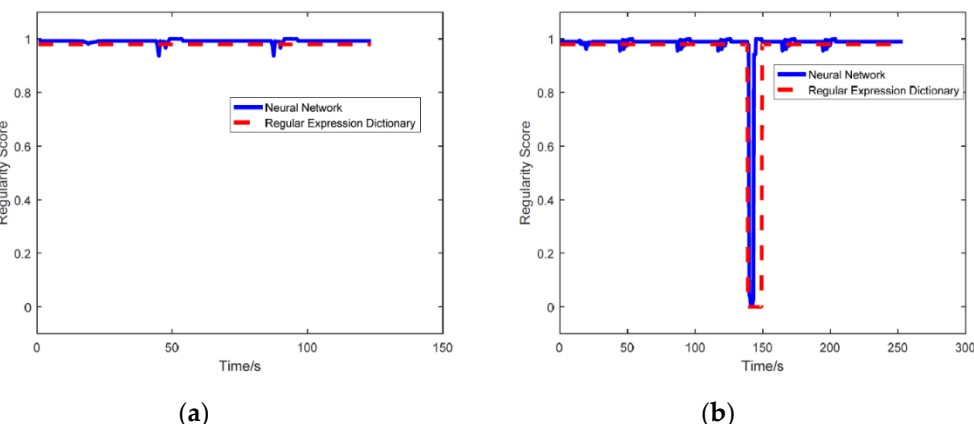

**Figure 7.** (**a**) Assessment for the normal vehicle. (**b**) Assessment for the abnormal vehicle.

For improved situational awareness of swarms of unmanned aerospace vehicles, a novel behaviour recognition algorithm was developed based on regular dictionary techniques (RD) to detect target behaviours. Simulations using multiple agents in a swarm have shown that the novel RD dictionary algorithm saves approximate 50% time than that of neural network methods when used in a swarm to detect multiple targets thus reduce the computational time needed.

## 5. Sensor Fusion

Sensor fusion is synonymous with technology to support the commander's situational awareness. Automatic target detection and tracking are fundamental sensor fusion tasks upon which both situational awareness and other support functions such as anomaly detection rely. An efficient system will detect targets early and produce reliable target coordinates, regardless environmental factors and hostile attempts to mislead or disrupt the system. Target tracking is not only concerned with the progress of target coordinates, but also the coordinates of mobile or UAV borne sensors that provide target input. A challenge in systems that rely on sensors swarms is indeed to and a (cost) effective solution that produce the sensor positions and orientation, collectively termed their pose. To that end, tracking also relies on sensor calibration, that is, the sensor data need to be interpreted in accurate real-world geometrical terms. Techniques for sensor positioning include satellite positioning systems (GNSS), inertial measurement units, ultra-wideband transceivers, simultaneous localization and mapping, and so on. These techniques will however, not be reviewed here. Instead, the focus will be on the relation between pose error and tracking feasibility and accuracy in scenarios relevant for the swarm mission scenario shown in Figure 1. The algorithm developed for tracking follows a decentralized architecture such as the one depicted in Figure 8. In contrast to distributed variants, detections are collected at a possibly local fusion node, where exclusively the target state in terms of real-world coordinates is estimated. To be mentioned, today's research and development, however, strive toward distributed tracking algorithms with an aim to increase fault tolerance while simultaneously meeting bandwidth restrictions.

The sensor fusion algorithm used for the swarm mission analysed in this work is broken down in the four steps. First an abstract idea on how the targets and sensors shall move is formulated using concrete matrices with sampled object coordinates and sensor detections. A motion planning program uses a script that indicates the object speed along the associated spline, and between station points. For each station point, the script determines the speed, or maximum speed, to be aimed for to the next station point, and if the object shall pause, or if there is a rendezvous with another object, and so on (Figure 9). The motion planning program samples the object coordinates at desired

sampling frequency, for instance 10 Hz, and also generates soft transitions by imposing continuity requirements on the acceleration. A six-state tracking filter is used to track all agents and targets. Figure 9a shows the targets and sensor allocations from the airborne UAV-sensor field of view, while Figure 9b shows the full view of three UAVs flying over a single ground target and performing the required monitoring task with sensor fusion from three UAV sensors.

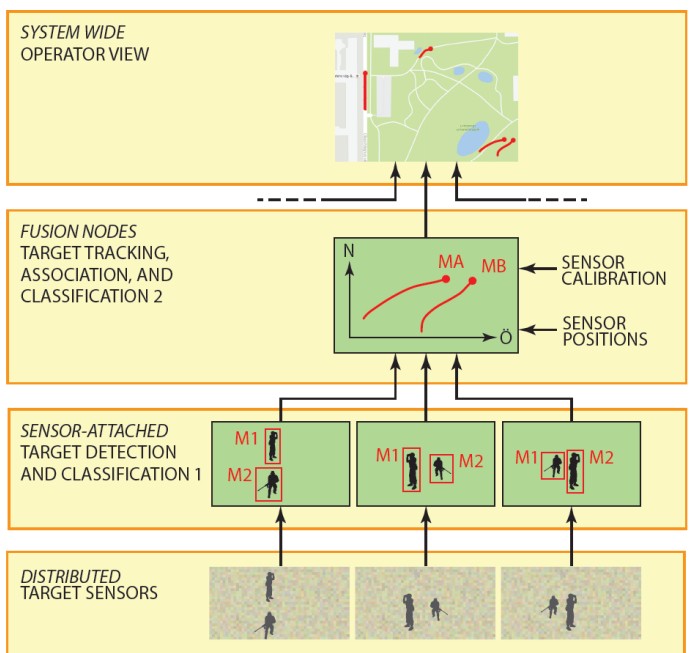

**Figure 8.** Breakdown of the target tracking components. Sensors were used with sensor-attached detectors that propagated target detections in terms of coordinates to a local fusion node.

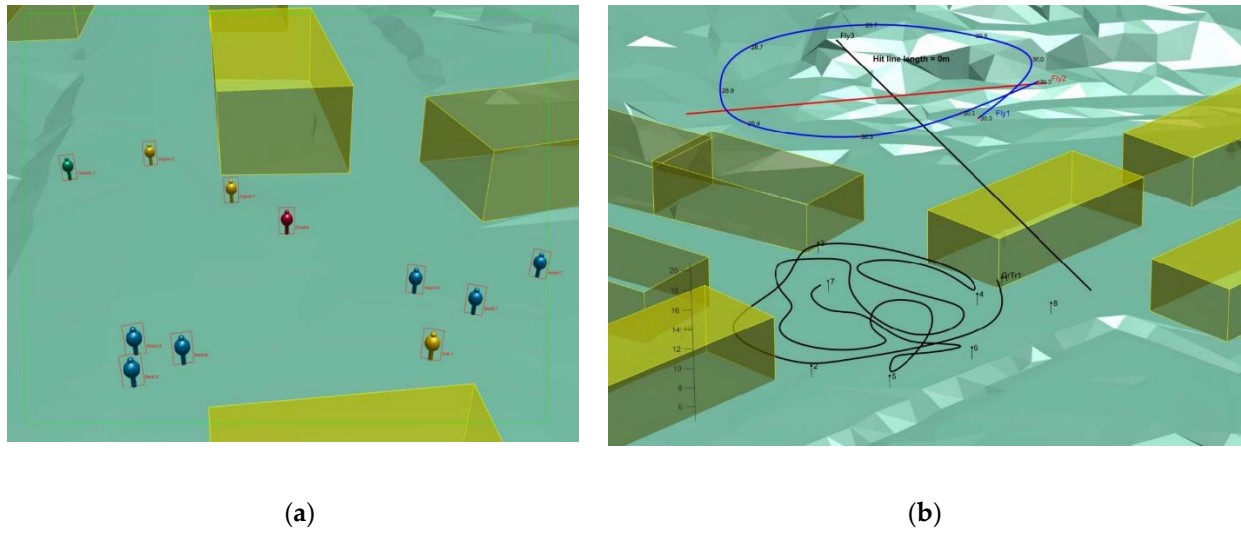

(**a**)          (**b**)

**Figure 9.** (**a**) Screenshot of simulated targets and sensor motions, where the sensor is a UAV-borne camera trained for the (red) person in the centre (**b**) full view.

The evaluation of the sensor fusion algorithm is based on Monte-Carlo simulations given a scenario with three UAV-born sensors over a single ground target, as shown in Figure 10. To evaluate the robustness against navigation errors, the true sensor position and orientation are perturbed by band limited Gaussian processes. The bandwidth of the perturbation is selected to 1/100 Hz. Regarding the orientation, the major perturbation

is on the yaw angle, while the pitch and yaw receive only a tenth of the variation. This reflects the assumption, that the yaw is more difficult to determine than the vertical axis, which can be given by a rather simple inclinometer or g-sensor. An ordinary magnetic compass, on the other hand, would be susceptible for spurious magnetic fields from power lines and iron masses and would also depend on local magnetic declination. Regarding the position, the perturbations for the east, north, and up components are independent and equally distributed. In the simulation, the ground target is moving at different speeds 3–8 km/h and is occasionally standing still at the station points. Three air borne sensors are trained at the ground target, of which two exhibit an (almost) perpendicular linear motion, and the third a circular motion. The sensor levels are 25–30 m above ground. The sensors continuously travel back and forth along their preferred trajectories with speed up to 10 km/h. No occlusion occurs in this scenario. However, the knowledge (certainty) of sensor positions and/or orientations is assumed to be limited. The simulation time is 185 s and the sampling frequency is 10 Hz. The primary evaluation criterion is the tracking root mean square error distance, RMSE.

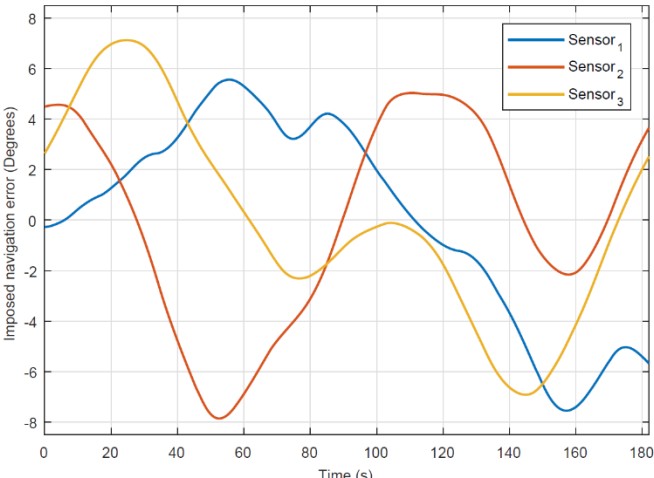

**Figure 10.** Simulated swarm navigation errors from UAV sensors.

Figures 11–14 show a single realization with and without collaborative positioning. Here, only the sensor orientation is perturbed with the realization illustrated in Figure 11, that is, $\sigma_{yaw} = 4°$ (standard deviation). The target tracking simulation outcome is presented in Figure 12 (without collaborative positioning) and in Figure 13 (with collaborative positioning.) Without collaborative positioning, to start with, the tracking error is on average RMSE-j = 1.2 m, but can apparently exceed 5 m, occasionally. Indeed, with collaborative positioning the error is significantly reduced, RMSE-j = 0.5 m, and the estimate follows the ground truth curve. For clarity it should be mentioned, that the RMSE is computed in three dimensions, while the vertical error is transparent in Figures 13 and 14. Collaborative positioning with yaw compensation allows larger orientation errors, but it has not always positive effect in the presence of position errors. Target tracking from a swarm of UAVs and UGVs requires that the platforms are fitted with suitable and well-calibrated sensors and that they have navigation ability to determine the position and orientation of the sensors with some accuracy. The tracking performance deteriorates quickly with increased navigation error. In the example scenario with three UAVs and a single ground target, the average yaw error must not exceed 4.5° and the average position error must fall below 0.4 m to meet an example requirement on tracking RMSE < 1 m. If the navigation units carry inertial measurement units, it can be assumed that the navigation error varies slowly, compared with platform and target movements. Then, techniques for collaborative positioning can be used to mitigate the effects of navigation errors, which in turn can reduce overall cost and weight on navigation solutions. In the example scenario it was demonstrated that collaborative positioning in the form of yaw error compensation reduced the susceptibility

for orientation error by almost 50%. It was also observed that yaw compensation alone can increase the sensitivity for position error, so future investigations of combined yaw and position error compensation are indeed warranted. In the swarm demonstration phase, practical test results with real time swarm-based tracking, based on the algorithm presented in this section will be presented, using real data from a swarm of unmanned aerial and ground vehicles.

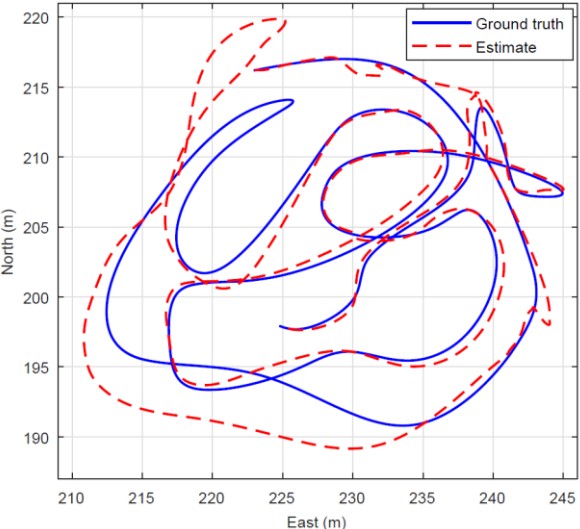

**Figure 11.** Simulation of target tracking. The continuous curve (blue) is the true track, and the dashed curve (red) is the EKF estimate. The corresponding tracking RMSE is 1.2 m.

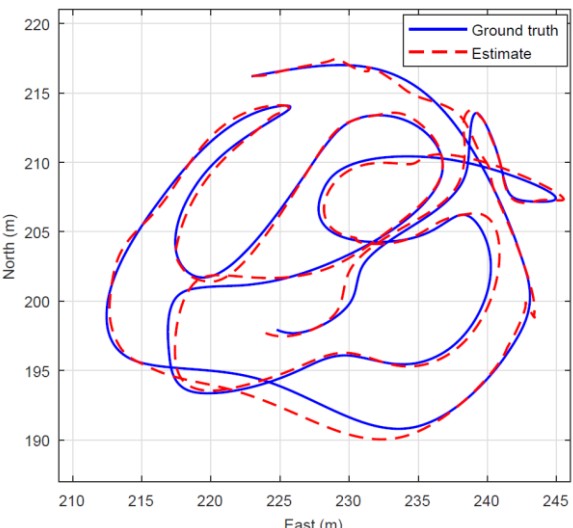

**Figure 12.** Simulation as in Figure 15 but with implementation of the collaborative positioning algorithm. The corresponding tracking is significantly reduced to <0.5 m.

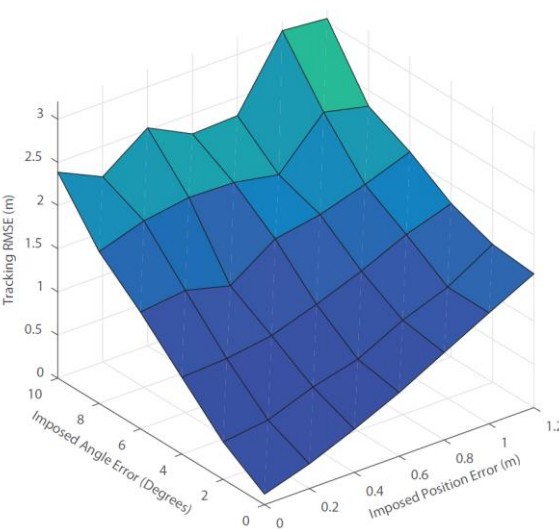

**Figure 13.** Tracking RMSE resulting from the Monte Carlo simulation.

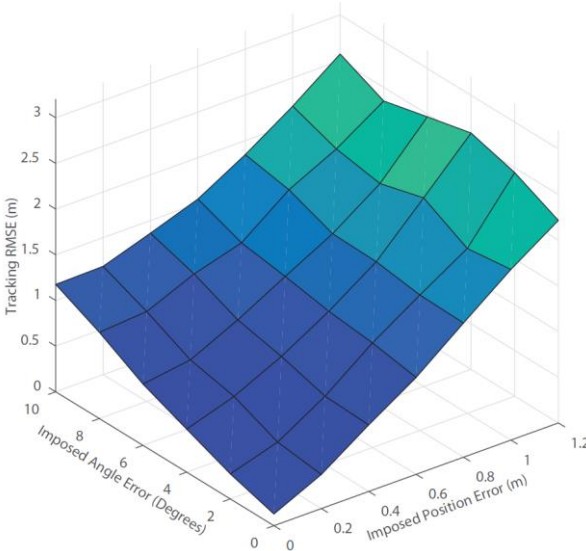

**Figure 14.** Tracking RMSE using a collaborative positioning algorithm. The corresponding tracking is significantly reduced by up to 50%.

## 6. Guidance for Swarm System Control

The mission of interest is the persistent monitoring mission described in Section 1. A static network of $N_g$ sensor is assumed to be deployed on the ground and provide information regarding to possible intruders inside an area of interest around a protected asset. In addition to the ground static sensors, $N$ mobile vehicles equipped with onboard sensors are assumed to be available. They may consist in ground mobile robots and/or aerial vehicles (UAVs). To define the objectives of the cooperative guidance and reconfiguration algorithms to be designed the following requirements are introduced and will be used:

- R1: enabling cooperation among the vehicles of the swarm to safely perform the monitoring task
- R2: ensuring complementarity between mobile vehicles of the swarm and the ground static sensor network
- R3: compensating by mobile vehicles for possible faults in the static sensor network.

From the mobile vehicles' point of view, the mission consists of reaching assigned targets while contributing to improve the monitoring of the area of interest along the

performed trajectories. The targets are assumed to be assigned by the "Mobile Vehicle Task Allocation" building block (see Figure 5) and defined as way points. They can correspond for example to locations of intruders detected by the ground sensors, to locations of areas over which complementary information is required to improve situational awareness or to locations of faulty ground sensors. It is assumed that a target is assigned to each vehicle of the swarm, with the possibility to assign multiple vehicles at the same location. In this section a reactive and distributed cooperative guidance law is designed as in Ref. [4] for the mobile vehicles involved in the analysed monitoring mission. A common criterion reflecting the mission and safety objectives is defined and evaluated according to each vehicle action and result, taking also into account the interaction between vehicles. Interactions between the vehicles and the static sensor network can also be handled by this criterion to ensure complementarity and reconfiguration in the monitoring mission. Guidance laws can be thus derived by optimization of this criterion, relying on approaches such as model predictive control (MPC).

### 6.1. UAV Swarm Guidance Algorithm

MPC has been widely used for the guidance of UAVs in various contexts, for example UAV flocking and formation flight have been discussed in [1]. In distributed MPC [5–7], each vehicle computes its control inputs at each timestep as a solution of an optimization problem over the future predicted trajectory. For tractability reasons, finite prediction and control horizon lengths, respectively, denoted as $H_p$ and $H_c$ are used. The future control inputs and the resulting state trajectories of a vehicle $i$ are written as:

$$U_i = \{u_i(t), u_i(t+1), \ldots, u_i(t+H_c-1)\}$$
$$X_i = \{\xi(t+1), \xi_i(t+2), \ldots, \xi_i(t+H_p)\} \tag{11}$$

If $H_c < H_p$, the control input is set to 0 after $H_c$ steps. Once the optimal input sequence $U_i^*$ has been computed, each vehicle communicates its predicted trajectory to the rest of the fleet and applies the first sample of the computed optimal control sequence $u_i^*(t)$. The optimization problems at time $t$ takes the following form:

$$\text{minimize } J_i(U_i, X_i)$$
$$\text{over } U_i \in U_i^{H_c} \tag{12}$$
$$\text{subject to } \forall k \in [t+1; t+H_p], \xi_i(k) \in X_i$$

where $J_i$ is the cost function associated with vehicle $i$. The constraints coupling the dynamics of the vehicles, such as collision avoidance, are taken into account by means of a penalty factor in the cost function. At the next timestep, each vehicle searches for its solution of the optimization problem. The cost function $J_i$ is composed of a weighted sum of terms reflecting the objectives of the mission. These terms are detailed in the following sections. Each cost function or its subcomponents are defined such that their norm is less or equal to 1 and weighted with a coefficient $w_\bullet$ to give priority to some of the objectives with respect to the others. Each vehicle defines its own trajectory online to achieve the mission objectives and constraints: (i) Head towards its assigned waypoint (ii) Maximize the cumulated area covered, in cooperation with the ground static sensors and the other mobile vehicles (iii) Avoid collisions between vehicles (iv) Minimize energy consumption to increase the monitoring capability. The associated global cost function is given as:

$$J_i = J_i^{nav} + J_i^{cov} + J_i^{safe} + J_i^{u} \tag{13}$$

The instant of time at which all these computations are carried out is $t$. The cost $J_i^{nav}$ to guide the vehicle $i$ towards its assigned waypoint $p_i^p$ has the following expression:

$$J_i^{nav} = \frac{1}{2H_p v_{max}} \sum_{n=t}^{t+H_p} w_p \|p_i(n) - \hat{p}_i(n)\| + w_f D\left(p_i(t+H_p) - B_i^{t+H_p}\right) \tag{14}$$

The first part penalises the distance of the predicted trajectory $p_i$ over the horizon $H_p$ to a virtual best-case trajectory $\hat{p}_i$ which is a straight line towards the waypoint $p_i^p$ at nominal speed $v_{nom}$. The second part encourages the vehicle to reach closer to the waypoint at the end of the predicted trajectory by penalising the distance to a reference ball $B_i$ around the waypoint as illustrated in Figure 15.

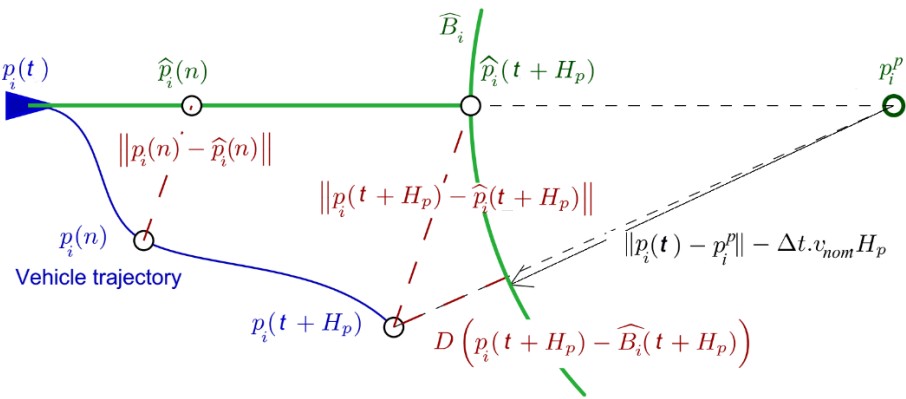

**Figure 15.** Definition of virtual trajectory and ball for navigation cost.

The cost $J_i^{safe}$ to avoid collisions between the vehicles is defined as:

$$J_i^{safe} = w_{safe} \cdot \frac{2}{H_p} \sum_{n=t+1}^{t+H_p} \sum_{\substack{j=1 \\ j \neq i}}^{N} \frac{1}{2}\left[1 + \tanh\left((d_{ij}(n) - \beta).\alpha\right)\right] \tag{15}$$

where $d_{ij}(n) = \|p_i(n) - p_j(n)\|$ denotes the distance between vehicles $i$ and $j$, and $p_i = [x_i, y_i]^T$.

The coefficients $\alpha$ and $\beta$, respectively, parameterize the width of the interval of fast variation of the hyperbolic tangent and its inflexion point. Two distances are defined: $d_{des}$ is the activation distance of the collision avoidance mechanism and $d_{safe}$ is the mandatory safety distance between vehicles. They are related to $\alpha$ and $\beta$ by $\alpha = \frac{6}{(d_{des} - d_{safe})}$ and $\beta = \frac{1}{2}\left(d_{des} + d_{safe}\right)$.

With this choice, the cost variation is less than 5% of its maximal value in the range $[d_{des}, +\infty]$. For implementation, the penalty function is set to 0 for $d_{ij} > d_{des}$, i.e., the vehicles do not consider each other above this distance. The cost $J_i^u$ to limit the energy spent by the vehicles is:

$$J_i^u = \frac{1}{H_p} \sum_{n=t}^{t+H_p} \frac{w_v}{v_{max}}(v_i(n) - v_{nom})^2 + \frac{w_\omega}{\omega_{max}}\omega_i^2(n) \tag{16}$$

It penalizes the difference between the actual speed and the desired speed $v_{nom}$ and favours straight lines over curved trajectories. The cost function $J_i^{cov}$ should reflect the gain in terms of map coverage for a potential trajectory. Each vehicle is assumed to have an attached sensor (of range $r_{sensor}$), described by a function $f_{cov}$ of the relative position between the observed point and the vehicle. The cooperatively covered area at time $t$ is:

$$\Omega = \bigcup_{\substack{n = 1,\ldots,t \\ i = 1,\ldots,N+N_g}} D_i^n \tag{17}$$

where $D_i^n$ is the sensing footprint of vehicle $i$ at timestep $n$. Since this representation is impractical, the mission field is approximated as a grid of resolution $d_{grid}$. A matrix $G$ stores the level of exploration of each cell of the grid. Each element $G_{l,m}$ (where $(l,m)$ are the integer coordinates of the cell of the grid) ranges between 0 when no vehicle has covered this location on a reference period and 1 when it has been entirely observed. Each vehicle stores a copy of this exploration map and updates it with the information from the rest of the fleet and the ground sensors (if their status is healthy). The precision of the representation only depends on the parameter $d_{grid}$. When a vehicle comes at a distance $d$ from the centre of cell $(l,m)$, the exploration level is updated:

$$G_{l,m}^+ = \max(G_{l,m}, f_{cov}(d)) \tag{18}$$

The exploration index is increased only if the vehicle is close enough. The function $f_{cov}$ is chosen to be continuous and identically 0 for $d > r_{sensor}$ as:

$$f_{cov}(d) = \begin{cases} 0 \ \ if \ d \geq r_{sensor} \\ \frac{1}{2}\left(1 + \cos\left(\frac{\pi d}{r_{sensor}}\right)\right) \ if \ d < r_{sensor} \end{cases} \tag{19}$$

The coverage matrix also takes into account the information from the ground sensor: the sensor footprint of each ground sensor whose status is known to be healthy is incorporated in $G$, hence ensuring complementarity between static and mobile sensors. In case of failure detection in a static sensor, the corresponding footprint is changed back to "not covered" in the map. This allows the mobile vehicles to take into account this new information in the computation of their next control input and naturally reconfigure the coverage mission of the swarm. The coverage cost function is thus defined to reward trajectories that cooperatively increase the global level of exploration of the map. Note that it takes negative values since the overall cost function is minimized, while the objective is to maximize the coverage.

$$J_i^{cov} = -w_{cov}\sum_{l,m} G_{l,m}(t + H_p) - G_{l,m}(t) = -w_{cov}\mathbf{1}^T\big(G(t + H_p) - G(t)\big)\mathbf{1} \tag{20}$$

where $G(t + H_p)$ is the predicted exploration map associated with the vehicle trajectory and $\mathbf{1}$ is the vector of appropriate dimension whose components are all $\mathbf{1}$. This cost function represents the total increase in the global coverage level resulting from a predicted trajectory. Since the vehicles share information, flying in already covered zones (by mobile or static sensors) is therefore, penalized and this also allows covering the area left free by a faulty sensor.

### 6.2. MPC Optimization and Cost Function Online Computation

The MPC optimization problem is a constrained nonlinear program, the solution of which cannot be found analytically. Numerical optimization must hence be used to approximate the solution. Global optimization procedures based, for example, on interval analysis [9] or genetic algorithms [10] can be used but may in practice be computationally prohibitive for real-time implementation. Numerical optimization methods, such as Sequential Quadratic Programming (SQP), Active Set or Interior Point methods are thus generally preferred [11]. Other methods suitable for MPC problems have also been developed [12,13]. Nevertheless, a global solution can be hard to find because of potential local minima. The computational time required for a MPC approach strongly depends on the parameterization of the control sequence. Low dimensional parameterizations have, for example, enabled successful applications to control systems with fast dynamics. Another solution consists of considering a finite set of predefined feasible control sequences, from which the one minimizing the cost function will be. This last solution is used in this paper for implementation of the MPC strategy, based on [8]. This systematic search strategy has several main advantages over a traditional optimization procedure. Firstly, the computation

load necessary to find a control sequence is constant in all situations leading to a constant computation delay. The second advantage is that the systematic search strategy can be less sensitive to local minima problems since the entire control space is explored. Finally, the systematic search requires no initialization of the optimization procedure. The studied search procedure consists of defining, prior to the mission, a set $S$ of candidate control sequences that satisfy control constraints. At each timestep, the control problem is solved using the proposed search procedure, as follows:

1. Using a model of the vehicle dynamics, predict the effect of each control sequence of the set of candidates $S$ on the state of the vehicle;
2. Remove from $S$ all of the candidate control sequences that lead to a violation of constraints on the state of the vehicle;
3. Compute the cost $J_i$ corresponding to each remaining candidate control sequence;
4. Select the control sequence that entails the smallest cost.

Since all of the feasible candidates in the set $S$ will be evaluated, the computation load of associated predictions should be as limited as possible. A simple parameterization of the control sequence is therefore, adopted, by considering a control input constant over the control horizon $H_c$ and then null over the remainder of the prediction horizon $H_p$. In addition, the distribution of the candidate control sequences is chosen so as to limit their number, while providing a good coverage of the control space.

The following three rules have been chosen:

- The set S of candidates includes the extreme control inputs, to exploit the full potential of the vehicles;
- The set S of candidates includes the null control input, to allow the same angular and linear accelerations to be continued with;
- Candidates are distributed over the entire control space, with an increased density around the null control input.

### 6.3. Swarm Guidance Algorithm Numerical Simulation

This section presents the evaluation and performance analysis of the proposed swarm guidance algorithm. Different simulation scenarios are considered to illustrate the compliance with the three requirements listed in Section 6.1 and influence of simulation parameters. The parameters of the swarm simulation scenarios performed are shown in Table 4.

**Table 4.** Swarm guidance simulation parameters.

| Parameter | Value |
|---|---|
| $(v_{min} v_{max} v_{nom})$ | $(0.3, 1, 0.7)$ ms$^{-1}$ |
| $(w_{min} w_{max} \Delta w_{max})$ | $(-0.2, 0.2, 0.05)$ rads$^{-1}$ |
| $(\Delta v_{max})$ | $0.1$ ms$^{-1}$ |
| $(d_{des} d_{safe} d_{grid})$ | $(9, 3, 2.5)$ m |
| $(w_p w_v w_{cov})$ | $1, 0.5. 2$ |
| $w_f w_\omega w_{safe}$ | $4, 0.5, 10$ |
| $r_{sensor}$ | $5$ m |
| $(H_p H_c)$ | $21, 3$ |

Five mobile vehicles represented by arrows are considered in Figure 16. The circle around each vehicle represents its sensor footprint. Five mobile vehicles are available. Two targets are assigned to two groups of two vehicles. A fifth target is assigned to the remaining vehicle. A fifth target is assigned to the remaining vehicle. A target way point (coloured dot) is assigned to each vehicle. The area covered by the sensor footprint of each vehicle during the mission is represented in grey levels corresponding to the exploration value (Equation (18)). Footprints of ground static sensors are represented by blue circles inside which the area is assumed to be covered. All vehicles successfully reach the assigned targets, starting from randomly chosen initial conditions in terms of position, orientation

and velocities. While arriving close to their targets, the vehicles start to perform trajectories to improve monitoring while staying as close as possible to the targets. This results in a quasi-circular trajectory for the green vehicle and local cooperative trajectories ensuring collision avoidance and complementarity in the monitoring for the three others. The ground static sensor network is assumed to be composed of:

- Four high range sensors ($r^g{}_{sensor}$ = 10 m) located at the centre of the area and with overlapping footprints (e.g., sensors monitoring a protected asset),
- Four middle range sensors ($r^g{}_{sensor}$ = 5 m) and 12 low range sensors ($r^g{}_{sensor}$ = 1.5 m) located all around the centre of the area as an "early warning frontier".

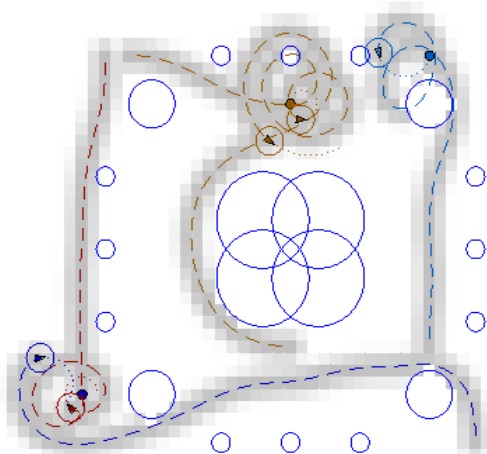

**Figure 16.** Monitoring mission—nominal scenario.

Illustration of fulfilment of the reconfiguration requirement (R3) is addressed through three simulation scenarios:

- a "nominal scenario" with no failure among the static sensors,
- a "reconfiguration scenario 1" with failure of two high range and one low range sensors during the execution of the mission,
- a "reconfiguration scenario 2" with failure of one middle range sensor and one low range sensor during the execution of the mission.

Faulty sensors are represented by red circles.

In the nominal scenario shown in Figure 16, cooperation between the vehicles enables to improve monitoring in a cooperative way to the static sensor network while minimizing overlaps as much as possible. As already mentioned, when the vehicles become close to their assigned targets, they keep moving close to the targets and improving coverage in this area. In the faulty scenario 1 presented in Figure 17, the trajectory of one of the vehicles is modified online to cope with failure of two high-range sensors. Some information is therefore, collected by opportunity over the areas not covered anymore by the static sensors. A failure has also been simulated for a small range sensor located on the top part of the area. Vehicles close to this sensor also start to modify their trajectories, because of the event. Online reconfiguration is also performed in the faulty scenario 2 shown in Figure 18, where trajectories of the vehicles are adapted online to compensate for missing information from the faulty sensors. Note that if one would wish to "permanently" compensate for faulty sensors by ensuring a full or persistent coverage of the area not monitored anymore, a new target should be assigned to some vehicle(s) in this/these location(s). Hence, the vehicle(s) would perform "circular"-like motions over the area(s) compensating for the faulty sensor(s).

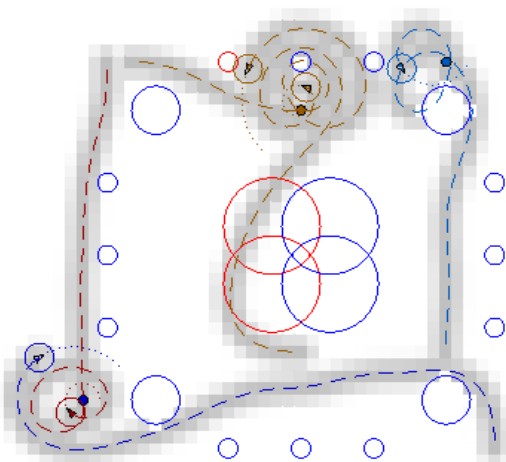

**Figure 17.** Monitoring mission—faulty scenario 1 (faulty sensors in red).

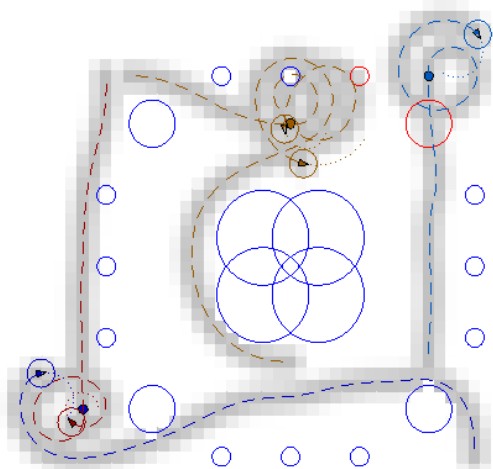

**Figure 18.** Monitoring mission—faulty scenario 2 (faulty sensors in red).

In summary, a reactive and distributed cooperative FDIR/guidance algorithm has been designed for mobile vehicles enabling cooperation and reconfiguration has been proposed for a persistent monitoring mission. The algorithm is based on Model Predictive Control and is designed in a distributed way, enabling each vehicle to compute its own control input, resulting in more robustness with respect to failure of one of the mobile vehicles. The algorithm enables to ensure monitoring complementarity among the vehicles in combination with a ground static sensor network. It also addresses safety issues by avoiding collisions between the vehicles in case of conflicting trajectories. The algorithm is validated via simulation and online reconfiguration is shown to cope with possible failures among the ground sensors.

## 7. Practical Swarm Demonstration

All swarm technologies presented in earlier sections are combined, simulated and then validated via a practical demonstration in a scaled outdoor environment. Due to the time and budget constraints the testing areas was limited in size and the unmanned platforms used to form a swarm of robots were based on COTS systems available in the market. The main objectives of the practical demonstration were to set up the demonstration environment, communications network and test the swarm functionalities for a persistent monitoring scenario as shown in Figure 7. For the outdoor demonstration 8 air and ground unmanned vehicles/agents were used in combination with static and mobile (ground)

targets. All agents are targets and are connected to a WIFI network or used radio telemetry. The navigation information of each agent and targets are shared via ROS based protocols. The task allocation system also is connected to ROS and the results of task allocation are also shared in ROS. Each agent is using the allocated mission information which is shared by the task allocation algorithm. Multiple combinations of agents, type of vehicles and number of targets where implemented, thus assessing homogeneous (same type of vehicle/sensors) and heterogeneous vehicles (optical/IR sensors, fixed wing/quadrotor UAVs). The unmanned systems used, and their physical Specifications of agents are summarized in Table 5.

**Table 5.** Physical specification of agents.

| Parameter | Parrot Bebop 2 | Parrot Disco | Erle Copter | Erle Hexacopter | Erle-Rover |
|---|---|---|---|---|---|
| |  |  |  |  |  |
| Width | 330 mm | 1150 mm | 360 mm | 590 mm | 325 mm |
| Length | 330 mm | 580 mm | 360 mm | 515 mm | 465 mm |
| Height | 89 mm | 120 mm | 95 mm | 95 mm | 145 mm |
| Weight | 500 g | 750 g | 1300 g | 1700 g | 2100 g |
| Endurance | | | ~30 min | | |
| Max Speed | 40 km/h | 40 km/h | 30 km/h | 30 km/h | 5 km/h |

The targets consist of two types: static targets (Type 1) and moving vehicles (Type 2), both using the Erle-Rover UGV. The outdoor demonstration took place at the Cranfield University Airport on 8 February 2018. Considering the proposed mission scenario, resource availability, project time limitations and costs/vehicle compatibility, the following demonstration parameters are used: 4 quadrotors, 1 fixed wing micro drone and 3 unmanned ground vehicle (UGV), operating in a $100 \times 100 \times 30$ m volume using GPS-optical guidance/navigation as shown in Table 6.

**Table 6.** Unmanned vehicles for swarm outdoor demonstration.

| | Model | Type | Units |
|---|---|---|---|
| Agent | Bebop | UAV | 3 |
| Agent | Erle-copter | UAV | 1 |
| Agent | Rover | UGV | 1 |
| Agent | Disco | Fixed wing UAV | 1 |
| Target | Rover | UGV | 2 |
| Target | Stationary target | Stationary target | 5 |
| GCS | PC | GCS | 1 |

Figure 19a shows the guidance, navigation and control structure for the first outdoor demonstration. An Erle-copter and a rover are integrated as agents newly and they are operated with own navigation and control system in on-board system. The navigation systems are based on the GNSS and INS integrated navigation system. A fixed wing UAV (Disco) is operated as a top-layer observer.

Figure 19b shows the network structure for the swarm outdoor demonstration. All agents are targets are connected to ROS through WIFI network. The results of the target behaviour monitoring algorithm, target detection algorithm, and sensor fusion algorithm are shared in ROS. The target information which is acquired from these algorithms also can be utilized in task allocation algorithm through ROS. An example of one of the outdoor demonstration/trials is presented in Figure 20a, which took place at Cranfield University Airport on 9 February 2018. The swarm system consisted with four agents and three moving targets with the mission trajectory shown in Figure 20a, where the dotted lines represent moving target's trajectories, the square markers are the stationary targets, and the

solid lines are trajectories of each UAV. Figure 20b shows the target detection probability which indicates the probability of detection of all the targets. If the probability becomes 1, it means all the targets are detected. The results show that the maximum probabilities of detection converge to 1 in the heterogeneous swarm case.

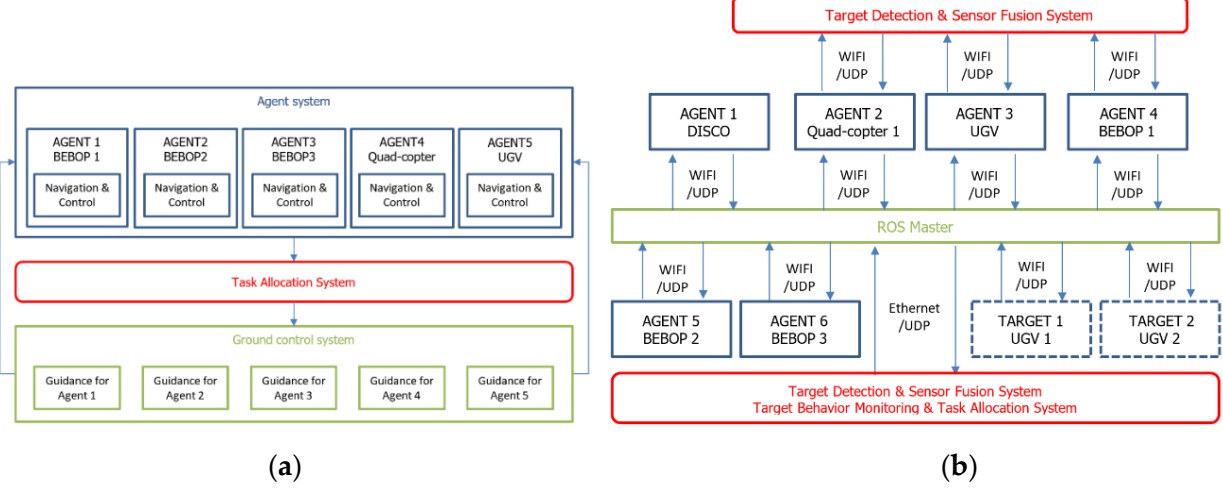

**Figure 19.** Swarm structure for (**a**) guidance, navigation and control system; (**b**) network.

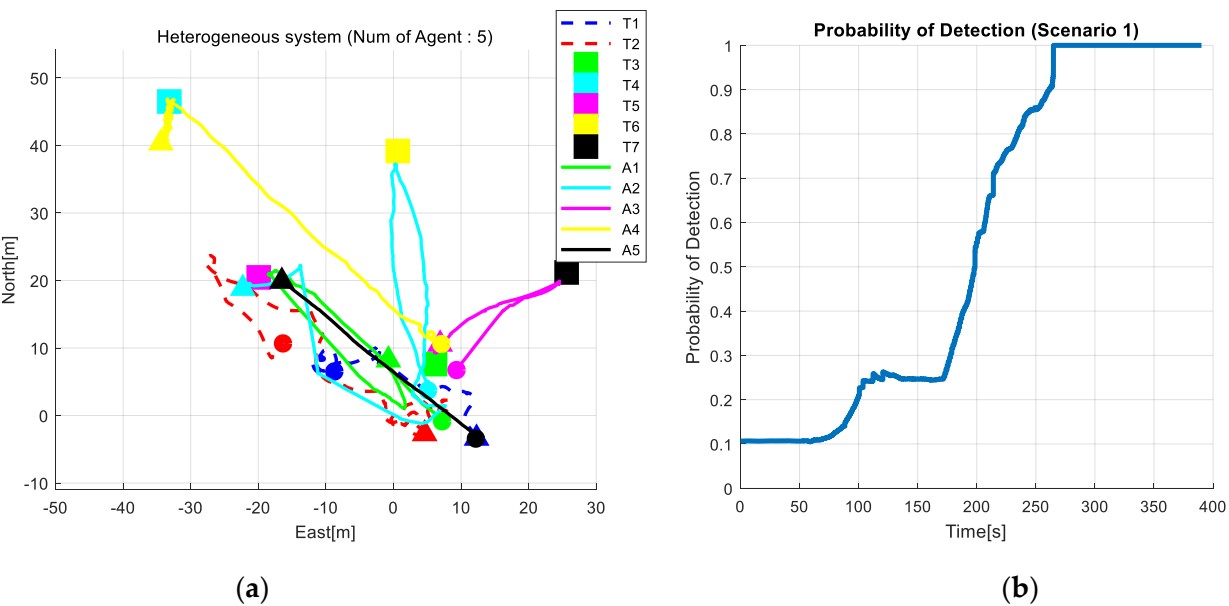

**Figure 20.** (**a**) Mission trajectory for the EuroSWARM outdoor experiment. (**b**) Time history of target detection probability.

## 8. Conclusions

A swarm of unmanned vehicles, mostly composed of micro-UAVs has been studied for use in the defence sector, for the protection of high-value assets such as military bases or installations. By testing and demonstrating an autonomous swarm of heterogenous vehicles, it has been shown that efficient and effective operation of unmanned swarm systems can bring a profound impact to the military arena. The key focus in the development of the enabling technologies has been the minimisation of uncertainties in situational awareness information for surveillance operations supported by a swarm 'system of systems' composed by static and mobile heterogeneous sensors. Critical enabling techniques and technologies for adaptive, informative and reconfigurable operations of unmanned swarm systems were developed in the work presented via the use of computationally efficient

algorithms for, mobile sensor tasking (including re-allocation), sensor fusion, information fusion including behaviour monitoring. Simulation and practical results, described in the paper, from a demonstration using a swarm of 10 micro-UAVs and UGVs has demonstrated the benefits of swarms of heterogeneous vehicles for defence applications such as for the persistent monitoring of high-value assets. Novel guidance, sensor fusion and task allocation algorithms which form the basic technologies for swarm systems have been matured in this work, through the development of algorithms which can be implemented in COTS based unmanned vehicles available today. The described algorithms have been integrated in a fully autonomous swarm system of a small scale and were designed and optimised to require small computational power, be flexible, reconfigurable and with the ability to be implemented in a large range of commercially available unmanned vehicles (air and ground). A realistic, persistent surveillance and monitoring scenario was implemented showing that efficient and effective operation of unmanned swarm systems can allow the swarm system end user in the battlefield to obtain real-time, relevant situational awareness information and help commanders make time efficient and effective decisions, while reducing risk/mission costs and human exposure to threats.

**Author Contributions:** Conceptualization, V.L. and A.T.; Formal analysis, D.L.; methodology, H.-S.S.; validation, S.B., J.M. and H.P.-L.; investigation, Y.D.; writing—review and editing, V.L. and H.-S.S.; project administration, V.K.; funding acquisition, V.L. All authors have read and agreed to the published version of the manuscript.

**Funding:** The work described in this paper was funded by the European Defence Agency through grant 'PP-15-INR-01—EuroSWARM: Unmanned Heterogeneous Swarm of Sensor Platforms'.

**Conflicts of Interest:** The authors declare no conflict of interest.

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
