# Peer review of "Autonomous Unmanned Heterogeneous Vehicles for Persistent Monitoring"

_drones, doi:10.3390/drones6040094_

Round 1
Reviewer 1 Report
Authors discuss swarm systems of the heterogeneous air & ground autonomous platforms for persistent monitoring applications. The manuscript presents an integration of swarm technologies for practical use. The integrated autonomous framework was designed and used in a scaled demonstration in an outdoor environment. The manuscript is well organized. To improve this paper for readers, I would like to suggest:
(page 5, line 187) Table 2 title should be moved on the same page as the table.
(General) The formulas have low quality (images?) -- please use an equation editor.
(page 8, lines 246 and 247) Add Table # and Figure #. In the submitted manuscript, it says "Table x" and "Figure x".
(General) Please check the location of figures within the text --- place figures immediately after they mentioned in the text first time if possible. Please check in the text that figures are correctly referenced. For example, page 16, line 515: Figure 9b is in the text, but there is only Figure 9, no (a) or (b)... Figure 10 is referenced in the text after Figures 11-14 --- please arrange the order for figures to make it clear to readers.
(page 19, line 599): "Gorecki1Error! Reference source not found" -- please fix it. I would like to suggest to use References listed at the end of the paper, not as footnotes. Similar error messages are in other sections of the paper (lines 609-611), and pages 21-22
(page 19, line 615): Equation (11) position on the page should be fixed. Similarly, equations (14), (15), (16), (19), (20)...
Author Response
We would like to thank the reviewer for their useful comments and feedback. These have been implemented in the revised version. Specifically:
Reviewer 1
Authors discuss swarm systems of the heterogeneous air & ground autonomous platforms for persistent monitoring applications. The manuscript presents an integration of swarm technologies for practical use. The integrated autonomous framework was designed and used in a scaled demonstration in an outdoor environment. The manuscript is well organized. To improve this paper for readers, I would like to suggest:
(page 5, line 187) Table 2 title should be moved on the same page as the table.
*Corrected
(General) The formulas have low quality (images?) -- please use an equation editor.
*Corrected (pdf conversion might make this worse)
(page 8, lines 246 and 247) Add Table # and Figure #. In the submitted manuscript, it says "Table x" and "Figure x".
*Corrected
(General) Please check the location of figures within the text --- place figures immediately after they mentioned in the text first time if possible. Please check in the text that figures are correctly referenced. For example, page 16, line 515: Figure 9b is in the text, but there is only Figure 9, no (a) or (b)... Figure 10 is referenced in the text after Figures 11-14 --- please arrange the order for figures to make it clear to readers.
*Corrected
(page 19, line 599): "Gorecki1Error! Reference source not found" -- please fix it. I would like to suggest to use References listed at the end of the paper, not as footnotes. Similar error messages are in other sections of the paper (lines 609-611), and pages 21-22
*Corrected
(page 19, line 615): Equation (11) position on the page should be fixed. Similarly, equations (14), (15), (16), (19), (20)...
*Corrected
Reviewer 2 Report
1. The paper lacks an in-depth analysis of current research in related fields. Trajectory estimation combined with neural networks is a popular research field. Some deep learning networks benefit from nonlinear modelings like "PFVAE: A Planar Flow-Based Variational Auto-Encoder Prediction Model for Time Series Data. Mathematics 2022, 10, 610. https://doi.org/10.3390/math10040610". 2. In section 4 of the paper, the advantages of the RD algorithm for abnormal vehicle detection compared with the neural network are insufficient, and there is no specific description of the neural network model used. 3. In the sensor fusion part of Section 5, the experimental basis for trajectory estimation is insufficient, lacking a maneuvering target tracking model and neural network experiment. Please discuss the differences and advantages, and disadvantages between the proposed method and the method combined with a neural network for trajectory estimation. 4. The formula in the article is not clear (lines 214 to 229). 5. The table in the article is not standardized enough (Table 2). 6. The serial number of the picture mentioned in the article does not correspond to the actual serial number.
Author Response
We would like to thank the reviewer for their comments and feedback. We have implemented most suggestions as requested:
Reviewer 2
- The paper lacks an in-depth analysis of current research in related fields. Trajectory estimation combined with neural networks is a popular research field. Some deep learning networks benefit from nonlinear modelings like "PFVAE: A Planar Flow-Based Variational Auto-Encoder Prediction Model for Time Series Data. Mathematics 2022, 10, 610. https://doi.org/10.3390/math10040610".
- The field of swarm related technologies is broad and there is indeed an extensive array of publications/solutions which can be used. Discussing these in depth would consume a lot of space and would be out of scope for our paper. However we have included a comment on line 91-93 and various analyses on some of the techniques used are presented at the beginning of each paragraph of our paper with references.
- In section 4 of the paper, the advantages of the RD algorithm for abnormal vehicle detection compared with the neural network are insufficient, and there is no specific description of the neural network model used.
- There has been text added on line 322 with references and Ref 16, 17 discuss the RD algorithm in detail. The advantages are clearly shown through the simulations presented (pages 12-14) and in lines 471-476 the algorithm shows a time saving of 50% compared to other neural network algorithms.
- In the sensor fusion part of Section 5, the experimental basis for trajectory estimation is insufficient, lacking a maneuvering target tracking model and neural network experiment. Please discuss the differences and advantages, and disadvantages between the proposed method and the method combined with a neural network for trajectory estimation.
- On page 18, lines 552-580 the advantages and disadvantages are presented including a Monte carlo simulation (Figure 13/14). The algorithm is able to reduce the orientation error by 50%. This is all further validated in the practical experiments (as in the RD algorithm above) in the following sections and thus we feel this clearly presents the advantages and disadvantages of the proposed method.
- The formula in the article is not clear (lines 214 to 229).
- Corrected
- The table in the article is not standardized enough (Table 2).
- We use a simple table to show the flow of the algorithms (and definition) for full clarity
- The serial number of the picture mentioned in the article does not correspond to the actual serial number.
- Corrected
Reviewer 3 Report
Authors have studied "Autonomous Unmanned Heterogeneous Vehicles for Persistent Monitoring" which is wonderful and novel work done.
Author Response
No changes required by the reviewer
Round 2
Reviewer 2 Report
There are also some text errors in the paper, such as the title of Table 4 is not on the same page as the table, the words in Table 5 are unclear, formulas (1-4) are unclear, and there are errors in line 701.